# Chiral higher spin gravity and convex geometry

**Alexey Sharapov[1], Evgeny Skvortsov[2] and Richard Van Dongen[2]**

**1** Physics Faculty, Tomsk State University, Lenin ave. 36, Tomsk 634050, Russia
**2** Service de Physique de l'Univers, Champs et Gravitation, Université de Mons,
20 place du Parc, 7000 Mons, Belgium

## Abstract

Chiral Higher Spin Gravity is the minimal extension of the graviton with propagating massless higher spin fields. It admits any value of the cosmological constant, including zero. Its existence implies that Chern–Simons vector models have closed subsectors and supports the $3d$ bosonization duality. In this letter, we explicitly construct an $A_\infty$-algebra that determines all interaction vertices of the theory. The algebra turns out to be of pre-Calabi–Yau type. The corresponding products, some of which originate from Shoikhet–Tsygan–Kontsevich formality, are given by integrals over the configuration space of convex polygons.

doi:10.21468/SciPostPhys.14.6.162

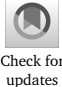
## 1 Introduction

The idea of Higher Spin Gravity (HiSGRA) is to construct quantum gravity models by exploring extensions of gravity with massless fields of all spins [1]. While masslessness can simulate the high energy regime, the importance of higher spin states for the quantum gravity problem is

supported by string theory, whose spectrum is populated by massive higher spin fields, and by the AdS/CFT correspondence since conformal field theories in $d \geq 3$ have single-trace operators of arbitrarily high spin, e.g. (Chern–Simons) vector models and $\mathcal{N} = 4$ SYM.[1] Even before constructing any HiSGRA it can be seen that the smallest multiplet is unbounded in spin with an $\infty$-dimensional symmetry behind [2–4].

Due to HiSGRA being at the brink it should not be surprising that constructing such theories faces numerous problems and naive attempts come immediately in contradiction with the basic field theory requirements [5–11]. Within AdS/CFT correspondence HiSGRA should be dual to vector models [12–15] or to (free) $\mathcal{N} = 4$ SYM [16–18], which is based on matching the spectrum of fields/operators. However, not every CFT has a well-defined, (i) quasi-classical, (ii) local dual with (iii) finitely-many fields. HiSGRA duals of vector models immediately violate (ii) and (iii). It is (ii) that presents a serious obstacle, eliminating the standard field theory tools in an attempt to construct the theories.[2]

With all hurdles already mentioned, it comes as a surprise that there exists a class of well-defined, local HiSGRA for any value of the cosmological constant, including zero. Its flat space origin is easier to explain. As is well-known [19, 20], there is a unique cubic amplitude/vertex for any three given helicities $\lambda_1 + \lambda_2 + \lambda_3 > 0$:

$$V_{\lambda_1,\lambda_2,\lambda_3} \sim [12]^{\lambda_1+\lambda_2-\lambda_3}[23]^{\lambda_2+\lambda_3-\lambda_1}[13]^{\lambda_1+\lambda_3-\lambda_2}, \tag{1}$$

and its complex conjugate is expressed in terms of $\langle ij \rangle$ and is valid for $\lambda_1 + \lambda_2 + \lambda_3 < 0$. Each theory comes with a particular spectrum of helicities and with cubic (and possibly higher) couplings $C_{\lambda_1,\lambda_2,\lambda_3}$,

$$V_3 = \sum_{\lambda_1,\lambda_2,\lambda_3} C_{\lambda_1,\lambda_2,\lambda_3} V_{\lambda_1,\lambda_2,\lambda_3}. \tag{2}$$

Chiral HiSGRA [21–23] is a unique class of theories that completes a genuine higher spin interaction[3] to a Lorentz-invariant, local theory. The spectrum has to contain all helicities (at least even) and the coupling constants are uniquely fixed to be [21–23]

$$C_{\lambda_1,\lambda_2,\lambda_3} = \frac{\kappa \, (l_p)^{\lambda_1+\lambda_2+\lambda_3-1}}{\Gamma(\lambda_1+\lambda_2+\lambda_3)}, \tag{3}$$

where $l_p$ is of length dimension. No higher order interactions are needed. All tree-level amplitudes can be shown to vanish on-shell (like in self-dual Yang–Mills theory and self-dual gravity) and the theory is at least one-loop finite [24–26]. These results were obtained in the light-cone gauge and argued to admit a smooth deformation to $(A)dS_4$, which was supported by [27, 28].

Chiral HiSGRA's spectrum suggests the dual to be (Chern–Simons) vector models. However, the interactions being chiral, it has to be dual to a closed subsector thereof [23, 28, 29]. Through the ABJ triality [30] one can also argue that its supersymmetric $\mathcal{N} = 6$ version is dual to a closed subsector of tensionless strings on $AdS_4 \times \mathbb{CP}^3$.[4] Chiral HiSGRA can also be detected through celestial studies [31, 32].

---

[1]There can be regions in the coupling space where, as in the SYM example, the higher-spin states decouple and the stress-tensor multiplet remains, but only in the first approximation.

[2]The results of [7–11] imply that this class of theories cannot be constructed by any Noether procedure. Nevertheless, one can 'reconstruct' them from correlation functions [8, 79, 80], which, however, has its own puzzles and is not applicable to Chern–Simons vector models before they are solved.

[3]One- and two-derivative Yang–Mills and gravitational vertices are not enough. One needs higher-derivative non-abelian interaction with a higher spin field.

[4]E.S. is grateful to Andre Coimbra for asking a question that leads to this idea.

The problem of covariantization and extension to $(A)dS_4$ of Chiral HiSGRA was solved in the recent papers [29,33,34]. However, the solution was wrapped into a rather abstract form of homological perturbation theory with considerable technical difficulties to extract explicit vertices and with many hidden gems that we uncover in this letter.

The main result of the letter is an explicit construction of classical field equations for Chiral HiSGRA. These originate from a cyclic $A_\infty$-algebra of pre-Calabi–Yau type [35]. The (higher) multiplications of the $A_\infty$-algebra are given by integrals over the configuration space of some convex polygons. This gives the first example of a local covariant HiSGRA with propagating massless fields. A grain of salt is that Chiral HiSGRA appears to be non-unitary due to interactions being complex in Minkowski signature and it is close in spirit to self-dual theories. Nevertheless, the theory should be unitary in flat space [24–26] at the price of having the trivial S-matrix, while in $(A)dS_4$, similarly to self-dual theories, the fact that it is a closed subsector of a unitary theory implies that all solutions and amplitudes of Chiral theory should carry over to the holographic dual of (Chern–Simons) vector models.

## 2 Chiral HiSGRA and $A_\infty$

### 2.1 Initial higher spin data

For a given set of physical degrees of freedom there is more than one way to incorporate them into Lorentz covariant (spin-)tensor fields. For a massless spin-$s$ particle a common way is to take a rank-$s$ symmetric tensor $\Phi_{\mu_1\cdots\mu_s}$, known as Fronsdal's field [36]. In the spinorial language its traceless component takes values in the $(s,s)$-representation of the Lorentz algebra,[5] $\Phi_{A(s),A'(s)}$. In principle, any spin-tensor $\Phi_{A(n),A'(m)}$ with $n+m=2s$ allows one to describe the same degrees of freedom. Most of these spin-tensor fields, while completely equivalent at the free level, resist including many important interactions, e.g. the gravitational one. For Chiral HiSGRA an optimal way to describe a massless spin-$s$ particle is rooted in the twistor approach [37–40]. Accordingly, positive and negative helicity states are placed into two different fields: the zero-form $\Psi^{A(2s)}$ and the one-form $\Omega^{A(2s-2)}$. The free action reads [41]

$$S = \int \Psi^{A(2s)} \wedge e_{AB'} \wedge e_A{}^{B'} \wedge \nabla\Omega_{A(2s-2)}\,, \tag{4}$$

where $e^{AA'}$ is the vierbein one-form and $\nabla$ is the compatible Lorentz-covariant derivative ($\nabla e^{AA'} = 0$). Note that $\nabla$ does not have to be flat/(A)dS and describes an arbitrary self-dual background, i.e., $\nabla^2\chi^A \equiv 0$ for any $\chi^A$. The action is invariant under the infinitesimal gauge transformations $\delta\Psi^{A(2s)} = 0$ and

$$\delta\Omega^{A(2s-2)} = \nabla\xi^{A(2s-2)} + e^A{}_{C'}\,\eta^{A(2s-3),C'}\,, \tag{5}$$

where the gauge parameters $\xi$'s and $\eta$'s are zero-forms. One can also append this action with Yang–Mills and gravitational interactions among the higher spin fields to construct higher spin extensions of self-dual Yang–Mills theory and of self-dual gravity [41,42]. These two theories are consistent truncations of Chiral HiSGRA [42], from which the scalar field can be safely discarded for the gravitational and Yang–Mills interactions are not much restrictive. Once a genuine higher spin interaction is present a unique completion gives Chiral HiSGRA.

---

[5]Hereinafter $A, B, \ldots = 1, 2$, $A', B', \ldots = 1, 2$ are the indices of two-fundamental representations of the Lorentz algebra, e.g. of $sl(2,\mathbb{C})$ for the Minkowski signature. We also abbreviate a group of totally symmetric (or to be symmetrized) indices $A_1 \ldots A_k$ as $A(k)$. All spinor indices are raised and lowered with the help of the anti-symmetric $\epsilon$-symbols $\epsilon_{AB}$ and $\epsilon_{A'B'}$ with $\epsilon_{12} = 1$.

The fields $\Omega^{A(2s-2)}$ can be packaged into a single gauge field $\Omega(y) = \sum_s \Omega^{A(2s-2)} y_A \cdots y_A$ assuming its values in the Weyl algebra $A_\lambda$. By definition, $A_\lambda$ is generated by the formal variables $\hat{y}_A$ subject to the canonical commutation relations $[\hat{y}_A, \hat{y}_B] = -2\lambda\,\epsilon_{AB}$. We prefer to think of the algebra $A_\lambda$ as resulting from the deformation quantization of $\mathbb{R}^2$, i.e., as the space of complex polynomials in $y_A$ endowed with the Moyal–Weyl $\star$-product. The numerical parameter $\lambda$ will be proportional to the square root $\sqrt{|\Lambda|}$ of the cosmological constant $\Lambda$. Likewise, after extending with the scalar field, the fields $\Psi^{A(2s)}$ can be packaged into a single matter field $\Psi(y)$. It is convenient to think of $\Psi(y)$ as taking values in the dual space $A_\lambda^* \simeq \mathbb{C}[[y_A]]$, the space of formal power series in $y$'s with complex coefficients. In the commutative/flat limit $\lambda = 0$, the Weyl algebra $A_\lambda$ passes smoothly to the commutative algebra of complex polynomials $A_0 = \mathbb{C}[y_A]$.

## 2.2 Sigma-model

The equations of motion of Chiral HiSGRA are constructed in the (AKSZ) sigma-model form:[6]

$$d\Phi = \sum_{n=2}^{\infty} l_n(\underbrace{\Phi, \ldots, \Phi}_{n}). \tag{6}$$

Here $\Phi$ is a collection of zero- and one-form fields, $d$ is the exterior differential, and the $l_n$'s are exterior polynomials in the form fields $\Phi$. The formal integrability of system (6), steming from $d^2 = 0$, imposes an infinite sequence of quadratic relations on the multi-linear functions $l_n$, which can be recognized as defining relations of a minimal $L_\infty$-algebra $\mathbb{L}$ [43]. System (6) admits a consistent truncation setting $l_n = 0$ for all $n > 2$. The remaining bilinear function $l_2$ defines then a graded Lie bracket on the target space of form fields $\Phi$. In general, the trilinear function $l_3$ is defined by a Chevalley–Eilenberg cocycle of the graded Lie algebra with bracket $l_2$.

In principle, any system of PDE can be cast into the form (6) at the expense of introducing auxiliary fields [44–46]. However, to describe propagating degrees of freedom the total number of components of the fields $\Phi$ must necessarily be infinite.

A remarkable fact is that the algebra $\mathbb{L}$ underlying Chiral HiSGRA originates from a minimal $A_\infty$-algebra $\hat{\mathbb{A}}$ via the standard symmetrization procedure.[7] The $A_\infty$-algebra $\hat{\mathbb{A}}$ decomposes further into the tensor product $\mathbb{A} \otimes B$ of a smaller $A_\infty$-algebra $\mathbb{A}$ and a unital associative algebra $B$. For Chiral HiSGRA $B$ is chosen to be $A_{\lambda=1} \otimes \mathrm{Mat}_N$. The factor $A_1$ is needed to have the right set of auxiliary fields for (6) to reproduce free equations of motion resulting from (4)[8] and $\mathrm{Mat}_N$ accommodates possible Yang–Mills gaugings. Since $\mathbb{A}$ and $B$ are completely independent of each other, we may keep the latter as an arbitrary parameter of $\hat{\mathbb{A}}$. It is also striking that the $A_\infty$-algebra $\mathbb{A}$ is of a very special type known as a pre-Calabi–Yau algebra of degree two [35]. This is defined as an $A_\infty$-algebra built on $A[-1] \oplus A^*$, where $A = \bigoplus A_n$ is a graded associative algebra, $A^*$ is its dual bimodule, and $A[-1]_n = A_{n-1}$. The $A_\infty$-products $m_n$ in $A[-1] \oplus A^*$ together with the natural pairing $\langle a|c\rangle = -\langle c|a\rangle$ for $a \in A$ and $c \in A^*$ give rise to the multi-linear forms $\langle m_n(\alpha_0, \ldots, \alpha_{n-1})|\alpha_n\rangle$ on $A[-1] \oplus A^*$ that are required to have cyclic symmetry. In our case, $\mathbb{A}$ is concentrated in degrees 0 and 1, i.e., $\mathbb{A} = \mathbb{A}_0 \oplus \mathbb{A}_1$, where $\mathbb{A}_0 = A_\lambda^*$ and $\mathbb{A}_1 = A_\lambda[-1]$. All the products $m_n$ in $\mathbb{A}$ are of degree $-1$. The canonical projection of

---

[6]It was first introduced by Sullivan [81] as a Free Differential Algebra and later leaked into supergravity [82,83] and higher spins [84]. In modern terms this is AKSZ equations of motion [85], see [86] for the relation to the HiSGRA problem and [46] for a broader context.

[7]This is just an $A_\infty$-extension of the familiar statement that the commutator in any associative algebra defines a Lie bracket.

[8]This set of auxiliary fields depends on the free spectrum and, for that reason, is exactly the same as in [84,87]. Some further simple projections/reality conditions may be needed, e.g. to reduce $\mathrm{Mat}_N$ to $U(N)$. An additional factor of Clifford algebra will lead to supersymmetric extensions.

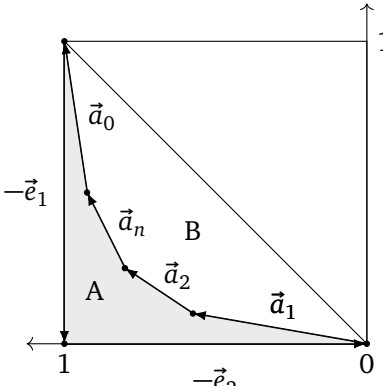

Figure 1: Configuration space: a convex polygon B and a swallowtail A. The swallowtail is built on $-\vec{e}_1, -\vec{e}_2, \vec{a}_1, \ldots, \vec{a}_n, \vec{a}_0$.

the field $\Phi$ onto the $A_\infty$-subalgebra $\mathbb{A} \subset \hat{\mathbb{A}}$ reproduces the pair of form fields $\Omega$ and $\Psi$ that we started with.

## 2.3 Configuration space and Convex Geometry

The products $m_n$ in the $A_\infty$-algebra $\mathbb{A}$ are reminiscent of (Shoikhet–Tsygan)–Kontsevich formality [47, 48]: they are poly-differential operators defined through integrals over configuration spaces of points inside and on the boundary of disk. Let us, however, stress that an extension of the formality that would generate the vertices of Chiral theory has not yet been identified. Nevertheless, one would expect that since the Poisson structure behind the Weyl algebra $A_\lambda$ is $\epsilon_{AB}$, i.e., constant and symplectic, the configuration space lacks the bulk part and reduces to points on the boundary. Therefore, the configuration space defined below should be identified with the boundary part of a yet to be found extension of the formality.

Our configuration space $\mathbb{V}_n$ admits two equivalent definitions: (a) the space of convex polygons that can be inscribed into a unit square and have one edge along the diagonal (region B in Fig.1) or (b) the space of all concave polygons with three convex vertices on the two adjacent sides of the unit square (region A), which we call swallowtails. Clearly, $\mathbb{V}_n$ is a compact region in $\mathbb{R}^{2n}$.

Both the definitions suffice for practical applications, but definition (b) can be improved. In a few words, one can consider the space of concave polygons with exactly three consecutive convex vertices on the oriented Euclidian plane $\mathbb{E}^2$ modulo the affine transformations of $GL^+(2, \mathbb{R}) \ltimes \mathbb{R}^2$. Any such polygon corresponds to a string of vectors $Q = (\vec{b}_1, \vec{b}_2, \vec{a}_1, \ldots, \vec{a}_n, \vec{a}_0)$ with $\vec{b}_{1,2}$ connecting the three convex vertices. Their affine coordinates form a $2 \times (n+3)$-matrix $Q$ with vanishing sum of columns (the polygon is closed). By $GL^+(2, \mathbb{R})$-transformations one can bring $Q$ into the canonical form

$$Q = \begin{pmatrix} -1 & 0 & u_1 & u_2 & \ldots & u_n & 1 - \sum u_i \\ 0 & -1 & v_1 & v_2 & \ldots & v_n & 1 - \sum v_i \end{pmatrix}, \tag{7}$$

with one of the convex vertices at the origin. Let $\boldsymbol{q}_{ij}$ denote the determinant of a $2 \times 2$ minor of $Q$ built from columns $i$ and $j$. Then the sum $|Q| = \sum_{i<j} \boldsymbol{q}_{ij}$ is equal to twice the area of the swallowtail A. $|Q|$ depends on $Q$ modulo cyclic permutations of the columns.

The configuration space $\mathbb{V}_n$ can also be understood as a subset in the oriented Grassmannian $\widetilde{\mathrm{Gr}}(2, n)$, with $\boldsymbol{q}_{ij}$ being the corresponding Plücker's coordinates [49]. It is not a positive one [50, 51], but the signs of $\boldsymbol{q}_{ij}$ are fixed.

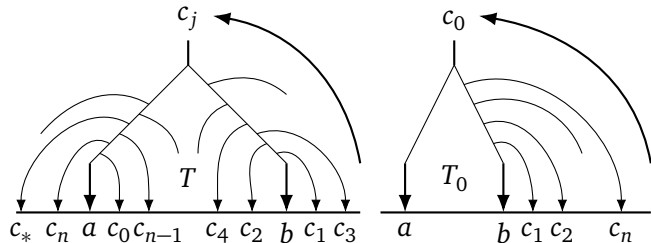

Figure 2: A generic tree $T$ on the left and the only tree $T_0$ contributing to (11) on the right. The counterclockwise arrow orders the arguments and $i$ of $c_i$ on the left panel corresponds to its position on $T_0$ as obtained via flips and shifts.

## 2.4 $A_\infty$-maps/Vertices

Now we are ready to construct the (higher) products $m_n(\alpha_1, \ldots, \alpha_n)$ for $\alpha_i \in \mathbb{A}$. As mentioned above they are given by poly-differential operators

$$m_n(p_0, p_1, \ldots, p_n) \alpha_1(y_1) \cdots \alpha_n(y_n)\Big|_{y_i=0}, \tag{8}$$

where $p_0 \equiv y$ and $p_i \equiv \partial_{y_i}$. The products respect $Sp(2)$-symmetry and are functions of the $Sp(2)$-invariant scalar products $p_{ij} \equiv -\epsilon_{AB} p_i^A p_j^B$ such that $\exp[p_{0i}] f(y_i) = f(y + y_i)$ is the shift operator. In particular, the pairing between elements $a \in A_\lambda$ and $c \in A_\lambda^*$ is defined as

$$\langle a|c\rangle = -\langle c|a\rangle = e^{p_{12}} a(y_1) c(y_2)|_{y_i=0}. \tag{9}$$

Thanks to the cyclicity of the $A_\infty$-structure on $\mathbb{A}$ many products are related to each other. At order $n$ one has $1 + \lceil \frac{n-1}{2} \rceil$ independent products. By dimensional considerations, each product $m_n$ may have either one or two arguments of $\mathbb{A}_1$. By cyclicity, the products with one argument are expressed through the products with two arguments. Therefore, we describe only those products $m_{n+2}$ that have two arguments $a, b$ of $\mathbb{A}_1$ and the other $n$ arguments $c_1, \ldots, c_n$, of $\mathbb{A}_0$. Such products are represented by sums over all planar rooted trees with exactly two branches, see Fig. 2. Each branch ends by one of the two arguments $a, b$. The other leaves are decorated by the arguments $c_i$. It is also convenient to decorate the root of the tree with an additional argument $c_0$, which corresponds to the scalar $\langle m_{n+2}(c_1, \ldots, a, \ldots, b, \ldots, c_n)|c_0\rangle$.

Let us start with the simplest tree $T_0$, see the right panel in Fig. 2. With each $c_i$ and $c_0$ we associate two vectors: $\vec{a}_i = (u_i, v_i)$, $i = 1, \ldots, n$, and $\vec{a}_0 = (1 - \sum_i u_i, 1 - \sum_i v_i)$, so that $\sum_i \vec{a}_i = 0$; $\vec{r}_i = (p_{a,i}, p_{b,i})$, $i = 1, \ldots, n$, and $\vec{r}_0 = (p_{0,a}, p_{0,b})$. This notation corresponds to the symbol of an operator that acts on

$$a(y_a) b(y_b) c_1(y_1) \cdots c_n(y_n)|_{y_\bullet=0}. \tag{10}$$

We also need an auxiliary matrix $P = (\vec{0}, \vec{0}, \vec{r}_1, \ldots, \vec{r}_n, \vec{r}_0)$ and recall that $Q = (-\vec{e}_1, -\vec{e}_2, \vec{a}_1, \ldots, \vec{a}_n, \vec{a}_0)$.

The tree $T_0$ describes a single contribution to the product $m_{n+2}(a, b, c_1, \ldots, c_n)$ with $a, b \in \mathbb{A}_1$ and $c_i \in \mathbb{A}_0$. The corresponding symbol reads

$$m_{n+2} = (p_{a,b})^n \int_{\mathbb{V}_n} \exp\left(\mathrm{tr}[PQ^t] + \lambda |Q| p_{a,b}\right), \tag{11}$$

see also (15). It generates many other structure maps via cyclicity. All other trees that contribute to $m_{n+2}$ give similar expressions. We just need to adjust $Q$ and $P$ in accordance with the topology of a given tree. Every tree $T$ can be obtained from $T_0$ by two operations: (1) flipping some $c_i$ from right to left on the right branch; (2) shifting the string of $c_1, \ldots, c_n, c_0$ by one unit

along the cord connecting $b$ to $a$. After these operations $c_i$ still remember their vectors $\vec{a}_i$ and $\vec{r}_i$ from $T_0$, e.g. $c_i$ of $T$ in Fig. 2 has $\vec{a}_i$ and $\vec{r}_i$ associated with it. They also remember their arguments, e.g. $c_i \equiv c_i(y_i)$ as it was for $T_0$. The arguments of the symbol that corresponds to $T$ should be read off from left to right. Therefore, the labeling of $c_i \equiv c_i(y_i)$, which is inherited from $T_0$, does not correspond to the natural labeling $c_1(y_1), \ldots, a(y_a), \ldots, b(y_b), \ldots, c_n(y_n)$. This will be compensated by a permutation $\sigma_T$ that reshuffles the labels on the $c_i$'s and $p_{ij}$'s. It also permutes the arguments $y_i$ of $c_i$. The symbol associated with $T$ consists of the sign, prefactor, principal term, and cosmological term (the last two are in the exponent). They are computed as follows.

*Sign.* For a given tree $T$ the sign $s_T$ is equal to $(-1)^m$, where $m$ is the total number of $c$'s in between $a$ and $b$. Note that this number is cyclic invariant, which is necessary for the cyclicity to work.

*Prefactor.* The prefactor is given by $(p_{a,b})^n$ where $n$ is the number of $c$'s; it acts on $a$ and $b$.

*Principal term.* To construct $P_T$ for a given tree $T$ one can just use the cyclicity applied to $T_0$. Explicitly, $P_T = (\vec{0}, \vec{0}, \vec{r}_1, \ldots, -\vec{r}_j, \ldots, \vec{r}_n, -\vec{r}_0)$, where $j$ is the $c_j$ at the root of $T$. The principal term is $\text{tr}[P_T Q^t]$.

*Cosmological term.* We associate the first two vectors $-e_1, -e_2$ of $Q$ (7) with $a$ and $b$. Recall that vectors $\vec{a}_i$ are associated with $c_i$, including $c_0$. In order to construct $Q_T$ for a given tree $T$ we fill in the columns of $Q_T$ starting from $a$ and then following the tree counterclockwise. The coefficient of $\lambda$ is $|Q_T| p_{a,b}$; it acts on $a$ and $b$.

Now we combine all the ingredients together and apply the permutation $\sigma_T$ to bring the labeling inherited from $T_0$ into the natural one. Thus, each tree $T$ makes the following contribution to $m_{n+2}$:

$$s_T \, \sigma_T (p_{a,b})^n \int_{\mathbb{V}_n} \exp\left(\text{tr}[P_T Q^t] + \lambda |Q_T| p_{a,b}\right). \tag{12}$$

One needs to sum over all trees $T$ to construct all $m_{n+2}$. We claim[9] that the products constructed in such a way do satisfy the defining relations of an $A_\infty$-algebra. Combining these products with the associative product in $B$, we can write the r.h.s. of equation (6) as

$$l_n(\Phi, \ldots, \Phi) = m_n(\Phi, \ldots, \Phi), \tag{13}$$

for the form field $\Phi$ with values in $\hat{\mathbb{A}} = \mathbb{A} \otimes B$. When restricted to the $\Phi$-diagonal, symmetrization is automatically performed, turning the $A_\infty$-algebra products into the multi-brackets $l_n$ of the $L_\infty$-algebra $\mathbb{L}$.

An important property of the poly-differential operators $m_n$ is that the corresponding symbols do not involve $p_{ij}$'s that connect $c_i$ with $c_j$. This translates into the locality of the vertices in the field theory language. Another important property is that the flat space vertices smoothly deform to $(A)dS_4$. In other words, the flat space limit is nonsingular.

## 2.5 Low order products

By way of illustrations let us present some low order products of $\mathbb{A}$. For $m_2$ the configuration space $\mathbb{V}_0$ is zero-dimensional and the corresponding swallowtail occupies half of the square (area $= 1/2$). The associated symbol

$$m_2(a, b) = \exp[p_{0,a} + p_{0,b} + \lambda \, p_{a,b}] \tag{14}$$

---

[9]This fact can be proven by extracting the $A_\infty$ products via homological perturbation theory of [29, 34]. However, the products obtained this way are very complicated and do not immediately reveal neither the relation to convex geometry nor to pre-Calabi–Yau algebras. The details of the derivation will be given elsewhere [88].

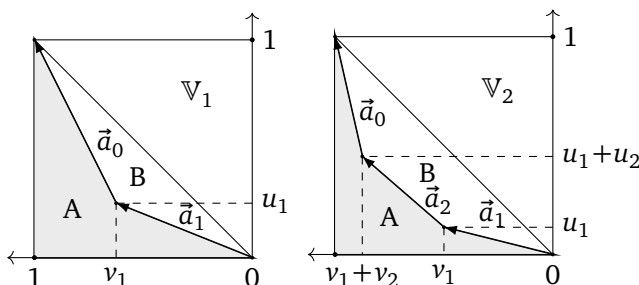

Figure 3: Configuration spaces $\mathbb{V}_1$ and $\mathbb{V}_2$.

is just the Moyal–Weyl $\star$-product with parameter $\lambda$. Cyclicity implies

$$\langle m_2(a,b)|c\rangle = \langle a|m_2(b,c)\rangle = -\langle b|m_2(c,a)\rangle \,,$$

which just gives the bimodule structure on $A_\lambda^*$, as designed. These vertices along reproduce most of the cubic amplitudes (1), (3) of Chiral HiSGRA [52].

For $m_3$ the space $\mathbb{V}_1$ is two-dimensional: one can place a point $(u,v)$ anywhere below the diagonal, the volume of $\mathbb{V}_1$ is $1/2$ and the area of A is $\frac{1}{2}(1+u-v)$, see Fig. 3. The duality determines 6 maps via just 2:

$$\langle m_3(a,b,c_1)|c_2\rangle = \langle m_3(c_2,a,b)|c_1\rangle \,,$$
$$\langle m_3(a,b,c_1)|c_2\rangle = \langle a|m_3(b,c_1,c_2)\rangle \,,$$
$$\langle m_3(c_1,a,b)|c_2\rangle = \langle m_3(c_2,c_1,a)|b\rangle \,,$$
$$\langle m_3(a,c_1,b)|c_2\rangle = \langle a|m_3(c_1,b,c_2)\rangle \,.$$

The two basis products can be chosen as $m_3(a,b,c)$ and $m_3(a,c,b)$, see (16).

For $m_4$ the space $\mathbb{V}_2$ is four-dimensional and its volume is equal to $1/24$. There are two independent products: one resulting from (15) for $n=2$ and another one given by the sum (17) over three trees.

## 3 Conclusions and Discussion

The main results of this letter are (i) an explicit construction of all covariant interactions vertices of Chiral HiSGRA both in flat and $(A)dS_4$ spaces; (ii) a remarkable relation between the vertices and convex polygons; (iii) a rich class of 2-pre-Calabi–Yau algebras $\hat{\mathbb{A}}$ that are parameterized by an associative algebra with trace. This gives the first example of a well-defined, local, manifestly covariant HiSGRA with propagating massless fields.[10] The results open up many obvious directions: (a) calculation of holographic correlation functions; (b) constructing exact solutions; (c) looking for an action [53] that would covariantize the light-cone results and extend them to $(A)dS_4$. Eventually, one expects Chiral HiSGRA to be integrable and UV finite, which is still to be proved.

In the regard to item (iii) a lot needs to be understood. It is clear that the field theory underlying Chiral HiSGRA is low dimensional since the functional dimension of $A_\lambda$ is two and

---

[10]Having well-defined vertices explicitly is very important. For example, [89] gives another type of homological perturbation theory in the same HiSGRA context, but the original recipe [89] to extract interactions leads to ill-defined vertices [90] (e.g. generic holographic correlation functions are infinite). From the field theory point of view [89] is not a concrete theory, but a general ansatz for interactions. It is not clear how to solve this problem since the duals of vector models are too nonlocal [7–11] to be treated by the standard tools. There are successful attempts to fix the first few local vertices (see [91,92] and refs therein) which, however, do not address the nonlocal ones where the actual problem resides.

the extra associative factor $B$ in $\hat{\mathbb{A}} = \mathbb{A} \otimes B$ is a passive spectator. This should be related to an important result of [42] that in the light-cone gauge the equations of motion can be cast into the form of the principal chiral model. Among other ideas, $\mathbb{A}$ can be used to construct field theories in $2d$ and $3d$. One can also construct a plenty of theories as a double-copy, just tensoring $\mathbb{A}$ with any $A_\infty$-algebra, e.g. $\mathbb{A} \otimes \mathbb{A}$.

Chiral HiSGRA is a local field theory in $AdS_4$ and it can be treated by the standard $AdS_4/CFT_3$-tools to give correlation functions of higher spin currents on the CFT side. They should cover a closed subsector of (Chern–Simons) vector models, which is yet to be identified. Nevertheless, the very existence of such a closed subsector supports [28, 54] the $3d$ bosonization duality [15, 55–59].

There is also a distant relation to tensionless string theory on $AdS_4 \times \mathbb{CP}^3$, which is via the ABJ triality [30]. One needs $\mathcal{N} = 6$ supersymmetric Chiral HiSGRA, which is easily achieved via the right $B$-factor of Clifford algebra [60]. Again, the very existence of Chiral HiSGRA implies that there should also exist a closed subsector of this tensionless string theory.

It is worth stressing that the 'formality' underlying the construction of Chiral HiSGRA is not yet known. The cubic product $m_3$ is equivalent [61, 62] to one that follows from the Shoikhet–Tsygan–Kontsevich formality [47, 48]. Higher structure maps are related to the deformation quantization of the Poisson orbifold $\mathbb{R}^2/\mathbb{Z}_2$ [62]. However, these relations can be seen at the formal $A_\infty$-level and do not give the specific vertices of the present paper. There should exist a topological field theory behind our construction [63], which the appearance of a pre-Calabi–Yau algebra also suggests [35].

The products of the $A_\infty$-algebra $\mathbb{A}$ underlying Chiral HiSGRA are fine-tuned to represent a local field theory. Indeed, a generic $A_\infty$-automorphism would lead to a non-local field redefinition for the vertices of sigma-model (6), thereby violating the equivalence theorem. This leads us to conclude that $A_\infty/L_\infty$-algebras representing actual field theories must have preferred bases where the vertices are maximally local. It would be interesting to reformulate the property of 'maximal locality' in a pure algebraic form.

Lastly, it is worth mentioning other interesting examples of HiSGRA: $3d$ topological [64–70]; $4d$ (and all even dimensions) conformal HiSGRA [71–73]; twistor constructions [53, 74–76]; IKKT-based [76–78]; holographic reconstruction [79, 80]. The last two examples relax the locality assumption in a controllable way and are not, strictly speaking, field theories. It seems even necessary to go beyond the field theory approach to construct HiSGRA's with massless fields that extend and complete Chiral HiSGRA.

$$m_{n+2}(a, b, c_1, \ldots, c_n) = (p_{a,b})^n \int_{\mathbb{V}_n} \exp\left[\left(1 - \sum_i u_i\right) p_{0,a} + \left(1 - \sum_i v_i\right) p_{0,b} + \sum_i u_i p_{a,i}\right.$$
$$\left. + \sum_i v_i p_{b,i} + \lambda \left(1 + \sum_i (u_i - v_i) + \sum_{i,j} u_i v_j \operatorname{sign}(j - i)\right) p_{a,b}\right].$$
(15)

$$m_3(a, b, c) = + p_{a,b} \int_{\mathbb{V}_1} \exp[(1 - u) p_{0,a} + (1 - v) p_{0,b} + u p_{a,1} + v p_{b,1} + \lambda(1 + u - v) p_{a,b}],$$

$$m_3(a, c, b) = - p_{a,b} \int_{\mathbb{V}_1} \exp[u p_{0,a} + v p_{0,b} + (1 - u) p_{a,1} - \lambda p_{a,b}(1 - u - v) - (1 - v) p_{b,1}]$$

$$- p_{a,b} \int_{\mathbb{V}_1} \exp[v p_{0,a} + u p_{0,b} + (1 - v) p_{a,1} - \lambda p_{a,b}(1 - u - v) - (1 - u) p_{b,1}].$$
(16)

$$m_4(a,c_1,b,c_2)$$

$$= -p_{a,b}^2 \int_{\mathbb{V}_2} \exp[(1-u_1-u_2)p_{0,a} + (1-v_1-v_2)p_{0,b} + u_2 p_{a,1} + \lambda A_1 p_{a,b} + u_1 p_{a,2} - v_2 p_{1,b} + v_1 p_{b,2}]$$

$$\quad - p_{a,b}^2 \int_{\mathbb{V}_2} \exp[(1-u_1-u_2)p_{0,a} + (1-v_1-v_2)p_{0,b} + u_1 p_{a,1} + \lambda A_2 p_{a,b} + u_2 p_{12} - v_1 p_{1,b} + v_2 p_{b,2}]$$

$$\quad - p_{a,b}^2 \int_{\mathbb{V}_2} \exp[u_2 p_{0,a} + v_2 p_{0,b} + (1-u_1-u_2)p_{a,1} - \lambda A_3 p_{a,b} + u_1 p_{a,2} - (1-v_1-v_2)p_{1,b} + v_1 p_{b,2}],$$

$$A_1 = 1 - u_1 v_2 + u_2 v_1 + u_1 - u_2 - v_1 - v_2, \quad A_2 = 2(1-v_1-v_2) - A_1, \quad A_3 = A_1 - 2u_1.$$

$$\tag{17}$$

## Acknowledgements

The work of E. S. and R. van D. was partially supported by the European Research Council (ERC) under the European Union's Horizon 2020 research and innovation programme (grant agreement No 101002551) and by the Fonds de la Recherche Scientifique — FNRS under Grant No. F.4544.21. A. Sh. gratefully acknowledges the financial support of the Foundation for the Advancement of Theoretical Physics and Mathematics "BASIS". E.S. expresses his gratitude to the Erwin Schrödinger Institute in Vienna for the hospitality during the program "Higher Structures and Field Theory" while this work was in progress. The results on the configuration space and Grassmannians were obtained under exclusive support of the Ministry of Science and Higher Education of the Russian Federation (project No. FSWM-2020-0033).

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
