# Peer review of "Chiral Higher Spin Gravity and Convex Geometry"

_SciPost Physics, doi:SciPost Phys. 14, 162 (2023)_

## Round 1 · Referee Report · Anonymous (Referee 1) · 2022-12-22

Strengths

1 - interesting results
2 - well-motivated
3 - concise

Weaknesses

The paper is:
1 - too concise to get a proper grasp of the technicalities
2 - just the short version of a (long) companion paper (arXiv:2209.15441) with complete technical details

Report

The goal of the paper is to report on the covariant form of the fully interacting chiral higher-spin gravity, achieved by the authors. The technical result is remarkable but the strategy is (by now) rather standard in the field : write the equations in AKSZ sigma-model and make use of formality / deformation-quantization techniques to address the problem.

The paper is written in a clear and intelligible way, the context of the problem and the achievements are well summarized. Moreover, the literature survey is quite extensive.

However, the letter format does not allow the authors to provide sufficient details so that arguments and derivations can be reproduced by experts. In fact, the technical details have appeared in a (long) companion paper (arXiv:2209.15441). Actually, the present paper is just the short version of the companion one. I cannot refrain mentioning that salami slicing scientific results on basically the same issue in five papers in the span of 5 months is a questionable practice, that should remain exceptional and not become standard in the future. As a referee, I am nevertheless inclined to make an exception in this case due to the considerable interest of the results.

In conclusion, I am in favor of recommending the paper for publication in SciPost if the authors take into acount the following suggestions.

Requested changes

1) The last sentence in the introduction ("This gives the first example of a local covariant HiSGRA with propagating massless fields.") is over-hyped and should be somewhat qualified because it could mislead non-experts. For instance, the authors should mention somewhere that chiral HiSGRA is non-unitary (even though it is expected to be a consistent subsector of a unitary HiSGRA). In fact, one should work either in Euclidean signature, or with complexified fields in the Lorentzian signature.

2) In the second paragraph of Subsection "Sigma-model", the authors use the notation A[1] without defining it (except if I missed something). This notation is standard in the DGA literature so I guess they have something like the suspension in mind, but they should properly define this notation in order to be self-contained.

3) The geometrical interpretation of the polydifferential operators is very neat and elegant. However, in the content of pages 3-4 it is not clear what is genuinely original and what follows (or closely inspired) from general techniques of formality / deformation-quantization applied to the specific example of pre-Calabi Yau A-infinity algebra at the bottom of p.2. This issue should be somewhat clarified by one (or two) short comment(s) somewhere. In fact, the originality of the construction has been clearly alluded to in the paragraph "It is worth stressing that (...)" in the conclusion. Nevertheless, this geometrical construction does not come out entirely from the blue, so a comment clarifying this point is in order.

4) In the conclusion, the authors should make precise the statement (i) (that is, "an explicit construction of all interactions vertices of Chiral HiSGRA") by stressing that the vertices are in covariant form (since, as explained, in the introduction, vertices were known in light-cone form where they stop at quartic order).

5) In the item [62], I suggest to slightly rephrase the comment "the original recipe (...) leads to nonsensical vertices" into a more balanced statement. For instance, by replacing the offensive term "nonsensical" by a more neutral expression like "divergent (so ill-defined)".

  • validity: top
  • significance: top
  • originality: good
  • clarity: high
  • formatting: excellent
  • grammar: good

Author:  Evgeny Skvortsov  on 2023-01-11  [id 3228]

(in reply to Report 1 on 2022-12-22)
Category:
correction

We would like to thank the Referee for endorsing the paper! We also would like to argue that the 'salami' technique has not been actually used, even though there have been quite a few papers on the covariantization of Chiral theory in 2022. While it was not even expected that the original Chiral theory in the light-cone gauge can be covariantized, the first paper gave a strong evidence in favor of this idea (all covariant cubic vertices were constructed). The second paper delivered all vertices for Chiral theory in flat space (via a homological perturbation theory, HPT). The third one extended this result to (A)dS. The success of each of these two steps was a great surprise and pure luck to some extent since the use of HPT does not solve any problem by itself. Moreover, getting something ill-defined via HPT is also very easy. Despite the fact that the HPT 'proves the concept', a lot of technical work is required to actually extract the vertices, which is what the last paper deals with. The present one capitalizes on a remarkable structure that the vertices have, which requires many additional steps as compared to the 'raw form' that comes out of the HPT. The papers have a very little overlap, except that we also summarize the main result of the present paper in the last one, to which we also refer for technicalities.

1) following the suggestion we expanded the discussion at the end of introduction to stress the drawbacks of Chiral theory. Accordingly, we now end with "A grain of salt is that Chiral HiSGRA appears to be non-unitary due to interactions being complex in Minkowski signature and it is close in spirit to self-dual theories. Nevertheless, the theory should be unitary in flat space \cite{Skvortsov:2018jea,Skvortsov:2020wtf,Skvortsov:2020gpn} at the price of having the trivial S-matrix, while in (A)dS4, similarly to self-dual theories, the fact that it is a closed subsector of a unitary theory implies that all solutions and amplitudes of Chiral theory should carry over to the holographic dual of (Chern--Simons) vector models"

2) We have added the definition of A[1], which is indeed the suspension and which now follows the most popular convention.

3) This is a very good and important point since it may appear that everything follows from Formality. Actually, the relation to Formality is just an analogy. None of the already available results on Formality can give the structure maps we have in the paper, save for the Moyal-Weyl star-product itself. We believe that there is an extension of Formality that would cover the case of Chiral theory and there are few hints towards this, which we discuss at the very end. We have rewritten the first paragraph of "Configuration space and Convex geometry" that ends with "Let us, however, stress that an extension of the formality that would generate the vertices of Chiral theory has not yet been identified. Nevertheless, one would expect that since the Poisson structure behind the Weyl algebra Aλ is ϵAB, i.e., constant and symplectic, the configuration space lacks the bulk part and reduces to points on the boundary. Therefore, the configuration space defined below should be identified with the boundary part of a yet to be found extension of the formality."

4) This is now fixed with "covariant vertices"

5) We have also changed "nonsensical" to "ill-defined", pointing out, as an example, that correlation functions are infinite.

---

## Round 1 · Referee Report · Anonymous (Referee 2) · 2023-1-13

Strengths

1- interesting problem
2- interesting results

Weaknesses

1- too technical
2- too concise

Report

The covariant formulation of chiral higher-spin gravity, the topic of this paper, is a very interesting problem. Even if this theory is somewhat limited due to its inherent chirality, the geometric / algebraic constructions associated to it may prove to be relevant more generally. Moreover, the theory may have a non-trivial parity-invariant completion, that could expand the framework of QFT in promising directions.

I broadly agree with the previous referee. The very nice construction introduced here is digestible for experts, but given the broader interest in the topic, a more pedagogical presentation would have been preferable, in my view. And yet, I agree that this is a significant contribution, and I recommend the paper for publication.

Requested changes

The authors already made helpful small changes following the previous report. I have two questions that I would like the authors to consider (not necessarily make changes).

1- The authors say below (3) that the theory is at least one-loop finite, but is this clear beyond the computations in [27-29] that were restricted to planarity?

2- The covariantisation introduced in this paper is at the level of the equations of motion. Does this extend to an action principle?

  • validity: top
  • significance: top
  • originality: high
  • clarity: good
  • formatting: excellent
  • grammar: good

Author:  Evgeny Skvortsov  on 2023-02-06  [id 3313]

(in reply to Report 2 on 2023-01-13)
Category:
answer to question

We would like to thank the Referee for approving the publication of the paper. Regarding the interesting questions, which are concerned with the properties of the chiral theory analyzed in other papers or general features of the formalism, we agree with the Referee that the answers, which we provide below, may not require any changes in the paper

1) the calculations of [27-29] are restricited to planarity or to the theory with all integer spins and no color. One quick argument for one-loop finiteness, which is not bound to planarity, made there is that the tree level amplitudes vanish and therefore the one-loop amplitudes cannot have nontrivial cuts. As a result there cannot be any log-terms signaling the usual log-divergences. The calculations were done in the light-cone gauge so far, which given the progress in covariantizing the theory, may not be pushed to higher orders and non-planar diagrams. In this regard, it is worth mentionning a recent e-Print: 2209.00925 by T.Tran where the Chern-Simons action on twistor space have been shown to capture correctly at least the cubic amplitudes. Something like Chern-Simons reformulation of the theory would explain finiteness to all orders.

2) we do not see any conceptual problems in writing down the action since within the perturbative expansion over flat/AdS space all (on-shell) vertices can be written in terms of the dynamical fields that are used in the equations of motion (here, the was a conceptual issue in the past that some of the higher spin interactions, e.g. the gravitational ones in flat space, cannot be written in terms of Fronsdal fields, but they can in terms of the chiral field variables used in the paper). However, writing down the action like this would be similar to expanding the gravity action in fluctuations, i.e. very tedious. Therefore, new techniques are needed to derive the proper (background independent) action. Again, it is worth mentionning 2209.00925 as a step towards the complete covariant action.

Anonymous on 2023-02-27  [id 3410]

(in reply to Evgeny Skvortsov on 2023-02-06 [id 3313])

Dear authors, my apologies for not replying to the previous response (not familiar with SciPost yet).

Thank you for the clarifying comments. I recommend that the paper is published as it stands.

Author:  Evgeny Skvortsov  on 2023-01-13  [id 3233]

(in reply to Report 2 on 2023-01-13)

We would like to thank the second Referee for endorsing the paper as well! The questions asked are very interesting and we provide some, hopefully satisfactory, answers below. However, the answers themselves go a bit outside the scope of the paper (related to the earlier results or to one of the future targets). Therefore, we hope that it is ok to provide the answers below.

1) The one-loop calculations of [27-29] apply either to the planar limit or to the chiral theory with all integer spins. In this latter case we are not restricted to the planar limit. Another argument is that the vanishing of tree level amplitudes should imply that there are no cuts at one-loop and, hence, no place for log-divergeces. There is also a very recent exciting result by Tran that (at least a part of) the action is captured by Chern-Simons theory on twistor space, which again points towards UV-finiteness.

2) It is exactly the fact that it was difficult to construct a covariant action (beyond the contractions that correspond to the higher spin extensions of self-dual Yang-Mills and gravity theories) that forced us to look for covariant equations of motion first. With this info available we know all interaction vertices on-shell and do not see any conceptual problems for the theory to have a covariant action (cf. Tran's result again), e.g. certain interactions do not have a local form in terms of Fronsdal fields, but this does not seem to happen with the chiral field variables. Therefore, we do not see any obstructions at the moment and the challenge is to find a neat form for this action (in all orders).

---

## Round 2 · Referee Report · Anonymous (Referee 1) · 2023-3-3

Strengths
1 - interesting results
2 - well-motivated
3 - concise
2 - well-motivated
3 - concise
Weaknesses
The paper is:
1 - too concise to get a proper grasp of the technicalities
2 - just the short version of a (long) companion paper (arXiv:2209.15441) with complete technical details
1 - too concise to get a proper grasp of the technicalities
2 - just the short version of a (long) companion paper (arXiv:2209.15441) with complete technical details
Report
The authors have taken my suggestions of changes into account and have answered the questions of the other referee, so I recommend the paper for publication in SciPost as it stands now.

Anonymous on 2023-03-03 [id 3429]
I recommend publication.

---

## Editorial Decision

published